# Catalytic [4+2]- and [4+4]-cycloaddition using furan-fused cyclobutanone as a privileged C4 synthon

Kemiao Hong[1,3], Mengting Liu[2,3], Lixin Qian[2], Ming Bao[1], Gang Chen[2], Xinyu Jiang[2], Jingjing Huang[2] & Xinfang Xu [1] ✉

Cycloaddition reactions play a pivotal role in synthetic chemistry for the direct assembly of cyclic architectures. However, hurdles remain for extending the C4 synthon to construct diverse heterocycles via programmable [4+n]-cycloaddition. Here we report an atom-economic and modular intermolecular cycloaddition using furan-fused cyclobutanones (FCBs) as a versatile C4 synthon. In contrast to the well-documented cycloaddition of benzocyclobutenones, this is a complementary version using FCB as a C4 reagent. It involves a C-C bond activation and cycloaddition sequence, including a Rh-catalyzed enantioselective [4 + 2]-cycloaddition with imines and an Au-catalyzed diastereoselective [4 + 4]-cycloaddition with anthranils. The obtained furan-fused lactams, which are pivotal motifs that present in many natural products, bioactive molecules, and materials, are inaccessible or difficult to prepare by other methods. Preliminary antitumor activity study indicates that **6e** and **6f** exhibit high anticancer potency against colon cancer cells (HCT-116, IC$_{50}$ = 0.50 ± 0.05 μM) and esophageal squamous cell carcinoma cells (KYSE-520, IC$_{50}$ = 0.89 ± 0.13 μM), respectively.

Furan-based heterocycles are ubiquitous in natural products, bioactive compounds, and functional materials[1-4], and they are also valuable synthetic building blocks that can be transformed into many other functional groups[5-7]. In particular, the furan-fused derivatives are widely used in the field of drug discovery[8-10]. Hence, the synthesis of these scaffolds with structural diversity is always an important topic in organic synthesis and has drawn much attention in the past decades[11-13]. Beyond the methods for the straightforward construction of furan-ring frameworks[14-19], a practical strategy through selective interception of in situ furan-based intermediates has been emerged as an effective and powerful method for the expeditious assembly of furan derivatives with fused architectures[20-24]. This cascade reaction protocol, in which multiple bonds form successively in a one-pot process and structural complexity is rapidly assembly, dramatically boosts the efficiency of synthetic endeavors. For example, the all carbon metal 1,3-dipole species **Int-1**, which is generated via transition-metal-mediated cyclization of yne-enones[25], has been well-documented as a versatile C3 synthon in the synthesis of 3,4-furan-fused molecules through a variety of [3+n]-cycloadditions (Fig. 1a, left)[25-30]. Furthermore, furan-based *o*-quinodimethanes (*o*-QDMs)[31,32] or azadienes[33] (**Int-2**) derived from cycloisomerization of enynones or enynamides, respectively, could be used for the efficient preparation of 2,3-furan-fused structures via [4+n]-cycloaddition reactions (Fig. 1a, middle). Despite these advances, the previously mentioned products are limited to only a few types of N- or O-heterocycles due to the structural uniqueness of these two furan-based species, **Int-1** and **Int-2**, for which their derivatization of which to synthesize lactams is very rare. The elegant discovery by Chi, involving a gold complex and NHC-carbene relay catalytic formal [4 + 2]-cycloaddition of ynamides and enals, represents the only example for the synthesis of furan-fused six-membered lactams[34]. In this context, the identification of furan-based intermediates in cycloaddition reactions is highly desired for the

[1]School of Chemistry and Chemical Engineering, Zhejiang Sci-Tech University, Hangzhou 310018, China. [2]School of Pharmaceutical Sciences, Sun Yat-sen University, Guangzhou, Guangdong 510006, China. [3]These authors contributed equally: Kemiao Hong, Mengting Liu. ✉e-mail: xuxinfang@zstu.edu.cn

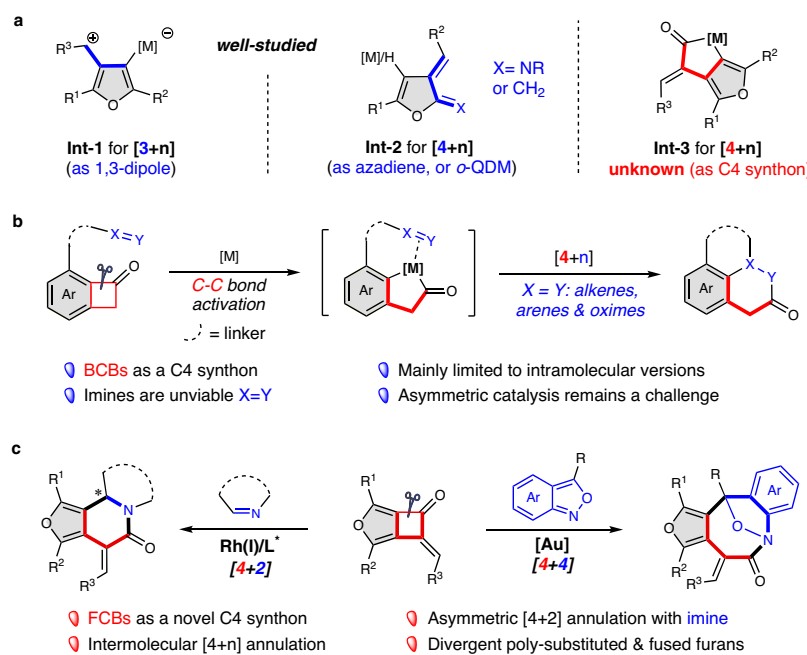

**Fig. 1 | Background and project synopsis. a** Reactive furan-based intermediates for the construction of poly-substituted & fused heterocycles. **b** [4+n]-Cycloaddition via C−C bond activation of benzocyclobutenones (BCBs). **c** Our strategy: [4 + 2]- & [4 + 4]-Cycloadditions via C−C bond activation of furan-fused cyclobutenones (FCBs).

construction of structurally diverse heterocycles as potential candidates for medicinal chemistry and drug discovery. For example, although only a few [4 + 4]-cycloadditions have been disclosed with prepared bench stable azadienes[35–42], an analogous version for the direct construction of furan-fused eight-membered lactams through interception of in situ formed C4 species remains elusive.

Recently, the transition metal (TM) catalyzed carbon-carbon (C−C) bond activation of benzocyclobutenones (BCBs), followed by intramolecular insertion of an unsaturated unit, has emerged as an attractive and atom-economical approach for the direct construction of fused-, spirocyclic-, and bridged-frameworks through [4+n]-cycloaddition (Fig. 1b)[43–47]. However, challenges remain in this area, including: (1) derivatization of BCBs to those that are heteroarene-fused remains elusive; (2) these transformations are mainly limited to the intramolecular versions due to the high reactivity of in situ formed metallacycle species; (3) the unsaturated units for [4 + 2]-cycloaddition have been primarily restricted to alkenes[48–51], alkynes[52–55], carbonyls[56], oximes[57], allenes/1,3-dienes[58], and heteroarenes[59], but imines and other readily available nucleophilic reagents are still unexploited; (4) only few catalytic asymmetric methods have been reported, and the asymmetric synthesis of chiral (hetero)arene fused heterocycles represents a significant challenge[60–65]. To address these limitations, the development of a platform molecule as a versatile C4 synthon is a pivotal aspect. Accordingly, with the development of corresponding catalytic C−C bond activation strategies, the in situ formed intermediate should exhibit distinctive chemical reactivities that facilitate unique types of bond formation that are difficult to construct with the use of conventional intermediates.

In our recent work, we have disclosed a dirhodium-catalyzed 4-exo-dig carbocyclization/[3 + 2] annulation cascade reactions of alkyne-tethered diazo compounds that provides a straightforward access to previously unknown furan-fused cyclobutanone (FCB) scaffolds[66]. Thus, we envisioned that these unique products might serve as a C4 synthon for the catalytic [4+n] annulations via C−C bond activation through metallocycle species Int-3 (Fig. 1a, right). Herein, we report our recent results in this direction, a modular intermolecular cycloaddition using FCBs as a versatile C4 synthon, including a Rh-catalyzed highly enantioselective [4 + 2]-cycloaddition with imines and

an Au-catalyzed diastereoselective [4 + 4]-cycloaddition with anthranils[67–69] (Fig. 1c). Notably, the former is a rare example of intermolecular asymmetric carboacylation of imines, and the latter is the only example of [4 + 4]-cycloaddition through interception of an in situ formed C4 intermediate for the direct construction of eight-membered lactams. Moreover, these furan-fused six/eight-membered polycyclic lactams, which are pivotal motifs present in many natural products, bioactive molecules, and materials, are inaccessible or difficult to prepare by other methods, even with multi-step precedures[70–72]. Furthermore, the derivatization of these resultant poly-functionalized fused molecules can be readily transformed into various heterocycles with structural diversity by routine manipulations, which demonstrates the significant synthetic potential of this protocol.

## Results

To test our hypothesis, we started our investigation by employing furan-fused cyclobutanone **1a** and cyclic imine **2a** as the model substrates. The desired furan-fused six-membered lactam **3a** could be obtained in 32% yield in the presence of Rh(COD)$_2$BF$_4$ (5.0 mol%) in DCE at 40 °C (entry 1, Table 1). This result prompted us to systematically optimize the reaction conditions to improve the yield of desired product **3a**, and the outcomes are summarized in Table 1. Other transition metals could also be used to catalyze this [4 + 2]-cycloaddition reaction (entries 2–10), with IPrAuNTf$_2$ giving higher yields (entry 8, 62% yield). Then, different solvents were screened with IPrAuNTf$_2$ as the metal catalyst (entries 11–16), showing that the isolated yields could be increased to 88% by conducting the reaction in MeCN (entry 15). Further increasing the reaction temperature to 60 °C did not provide better results (entry 17).

With the optimal reaction conditions in hand (Table 1, entry 15), the substrate scope with respect to imines **2** was investigated (Fig. 2, top). Remarkably, a variety of substituents at different positions of dibenzazepines were all well tolerated, delivering the [4 + 2]-cycloaddition products **3a**–**3n** in 78–95% yields. Moreover, the corresponding reactions proceeded smoothly when replacing the oxygen atom with a sulfur linkage, leading to the **3o** in 56% yield. Notably, phenanthridine displayed remarkable reactivity, giving **3p** in high yield. To our delight,

**Table 1 | Optimization of the reaction conditions[a]**

| Entry | Cat. | Solvent | Yield[b] (%) |
|---|---|---|---|
| 1 | Rh(COD)$_2$BF$_4$ | DCE | 32 |
| 2 | Pd$_2$(dba)$_3$ | DCE | ND |
| 3 | Ni(COD)$_2$ | DCE | NR |
| 4 | Cu(OTf)$_2$ | DCE | 15 |
| 5 | AgSbF$_6$ | DCE | 25 |
| 6 | [Ru($p$-cymene)Cl$_2$]$_2$ | DCE | NR |
| 7 | PPh$_3$AuNTf$_2$ | DCE | ND |
| 8 | IPrAuNTf$_2$ | DCE | 62 |
| 9 | BrettphosAuNTf$_2$ | DCE | 26 |
| 10 | JohnPhosAu(MeCN)SbF$_6$ | DCE | 56 |
| 11 | IPrAuNTf$_2$ | DCM | 50 |
| 12 | IPrAuNTf$_2$ | THF | NR |
| 13 | IPrAuNTf$_2$ | toluene | NR |
| 14 | IPrAuNTf$_2$ | PhCF$_3$ | 46 |
| 15 | IPrAuNTf$_2$ | MeCN | 89 (88)[c] |
| 16 | IPrAuNTf$_2$ | EtOAc | NR |
| 17[d] | IPrAuNTf$_2$ | MeCN | 78 |

*NR* no reaction, *DCE* dichloroethane, *DCM* dichloromethane, *THF* tetrahydrofuran.

[a]The reaction was carried out on a 0.1 mmol scale: to a solution of **1a** (37.9 mg, 0.12 mmol, 1.2 equiv.), and **2a** (19.5 mg, 0.1 mmol, 1.0 equiv.) in 1.0 mL solvent, was added a solution of catalyst (5.0 mol %) in 1.0 mL of same solvent via syringe under argon atmosphere at 40 °C, and the reaction mixture was stirred for 18 h under these conditions.

[b]Determined by ¹H NMR analysis of the crude reaction mixture based on internal standard (1,3,5-trimethoxybenzene).

[c]Result in parentheses is isolated yield.

[d]The reaction was conducted at 60 °C.

quinoxalinone afforded the **3q** in a synthetically useful yield. Finally, the other two cyclic ketimines, **2r** and **2s**, were also suitable substrates, providing the desired products **3r** and **3s** in 80% and 40% yields, respectively.

Subsequently, we investigated the scope of furan-fused cyclobutanones under standard conditions (Fig. 2, bottom). FCBs **1** bearing different functionalities, including chloro, bromo, fluoro, trifluoromethyl, cyanide, ethoxy, methyl, phenyl, and methoxy groups at different positions on the 5-aryl ring participated in this reaction to form the expected six-membered lactams **4a–4m** in 39–89% yields. Likewise, the 1-naphthyl substituted reactant underwent the reaction smoothly, delivering **4n** in 80% yield. Notably, alkyl halide is also a compatible substituent under these conditions, exemplified by the formation of the chlorine-containing product **4o** in 83% yield. Subsequently, the substitution pattern on the styryl moiety was examined, resulting in six-membered N-heterocycle **4p** in 78%. Notably, the variation of the ether part had a negligible influence on the reactivity, giving the bromoethyl derivative **4q** in 70% yield. In addition, the structure of product **4k** was confirmed by single-crystal X-ray diffraction analysis, and the stereochemistry of the others was assigned analogously.

Encouraged by the above results, we envisioned that higher-order cycloaddition using FCB as the C4 synthon might be feasible with appreciate dienophiles under compatible catalytic conditions. Thus, we further explored of the gold-catalyzed [4 + 4]-cycloaddition by using commercially available anthranil **5a** and FCB **1a** as model substrates[67]. After optimization, the best conditions were identified using JohnPhos(MeCN)AuSbF$_6$ (5.0 mol%) as the catalyst, DCE as the solvent at 40 °C for 18 h, affording the desired product **6a** in 93% yield

(see Supplementary Table 1 in the Supplementary Information for details). With these optimal reaction conditions in hand, the scope with respect to FCBs **1** was investigated (Fig. 3, top). A range of FCBs with aryl groups bearing different substitutions, such as chloride (**6b**), bromide (**6c**), fluoride (**6d, 6e, 6i**), methyl (**6f**), ethoxy (**6g**), and trifluoromethyl (**6h**) were all tolerated, providing the desired cycloadducts **6b–6h** in good to superior yields (79–92%). For FCBs **1** bearing 1-naphthyl, 1-thienyl, the corresponding annulation products **6j** and **6k** were obtained in 89% and 61% yields, respectively. To our delight, the corresponding reactions proceeded smoothly when the aryl ring was replaced with an alkyl halide, leading to **6l** in 60% yield. The substituents on the styrene motif (e.g., 2-methoxy, 4-bromide, and 4-trifluoromethyl) have little effect on the reaction outcomes, and the [4 + 4] annulation products **6m–6o** were obtained in 74–89% yields. Switching the ethyl group on the ether unit (R¹) with *tert*-butyl or benzyl was highly compatible, and the reactions occurred with high efficiency, leading to **6p–6q** with excellent results (85–93%). The *D*-menthol and estrone-derived furan-fused cyclobutanones also worked very well under optimal conditions, delivering **6r** and **6s** in 87% and 88% yields, respectively. In addition, this reaction could be conducted on a 1.5 mmol scale, providing 581 mg **6a** in 89% yield (note b). The structure of **6f** was established by X-ray crystallographic analysis, and the stereochemistry of the others was assigned analogously.

Next, the scope of anthranils **5** was examined with FCB **1a** (Fig. 3, bottom). Various functional groups, such as halides (F, Cl, Br), methoxy, methyl, methyl ester, and pivalate were compatible, affording the [4 + 4] cycloaddition products **7a–7j** in good to excellent yields. Among them, the sterically hindered C4-substituted and C7-

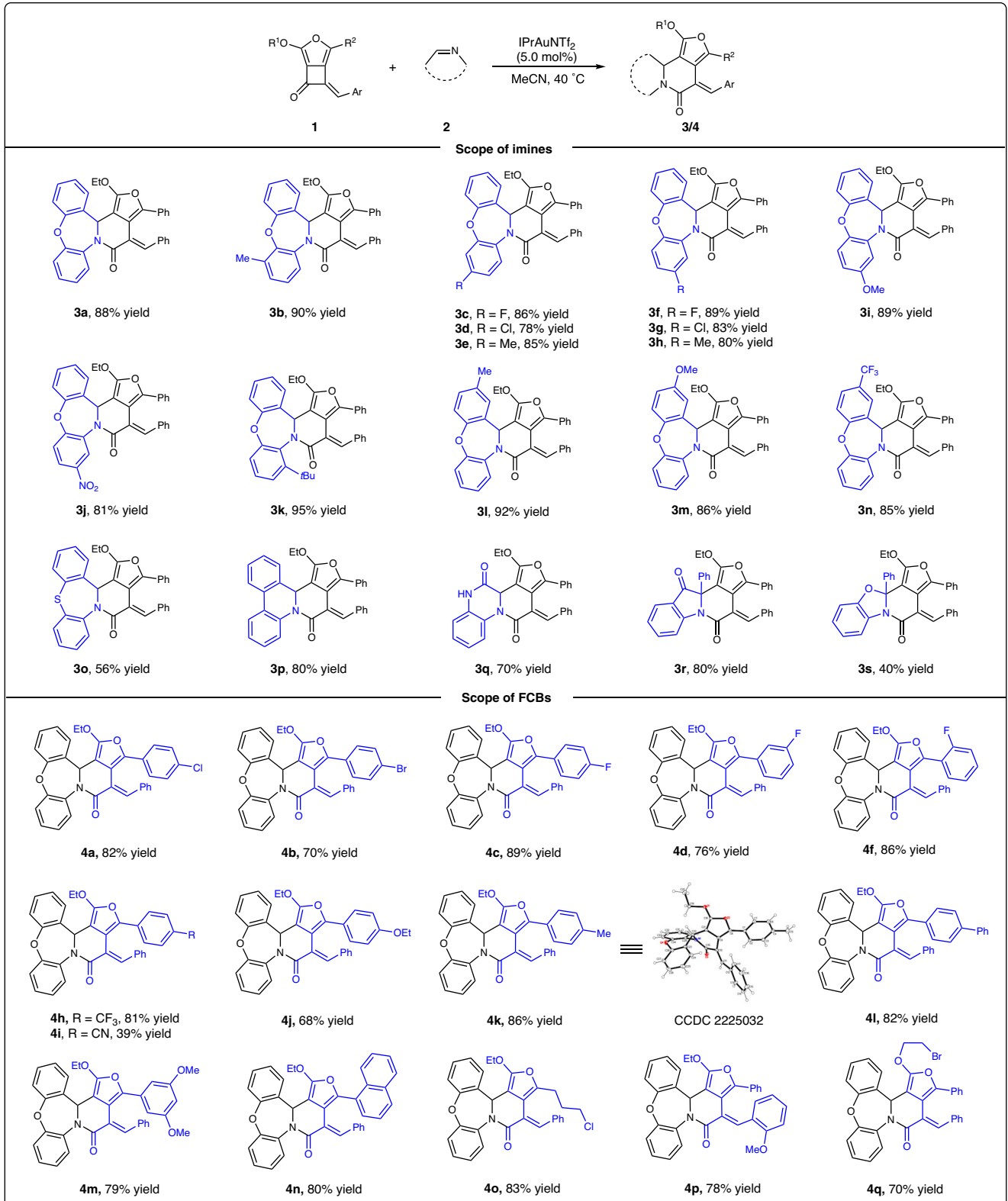

**Fig. 2 | Substrate scope for the [4 + 2]-cycloaddition.** The reaction was carried out on a 0.1 mmol scale: **1** (0.12 mmol, 1.2 equiv.), **2** (0.1 mmol, 1.0 equiv.), and IPrAuNTf$_2$ (4.3 mg, 5.0 mol%), MeCN (2.0 mL) was added via syringe under argon atmosphere at 40 °C, and the reaction mixture was stirred for 18 h under these conditions. Yields given are isolated yields.

substituted anthranils (**5a** and **5j**) also reacted well, yielding **7a** and **7j** in 76% and 91% yields, respectively. The electronic-rich substrate **5k** with the 5,6-dimethoxyl substitute, reacted smoothly to give **7k** in 83% yield. Furthermore, anthranils with methyl, ethyl, and phenyl groups at the C3 position afforded furan-fused eight-membered lactams **7l–7n** in excellent yields. Notably, the naproxen-derived anthranil also worked very smoothly under optimal conditions, generating the product **7o** in 86% yield as two diastereomers.

**Fig. 3 | Substrate scope for the [4 + 4]-cycloaddition.** The reaction was carried out on a 0.1 mmol scale: to a solution of **1** (0.1 mmol, 1.0 equiv.) and **5** (0.11 mmol, 1.1 equiv.) in 1.0 mL DCE, was added a solution of JohnPhosAu(MeCN)SbF$_6$ (3.9 mg, 5.0 mol%) in 1.0 mL DCE via syringe under argon atmosphere at 40 °C, and the reaction mixture was stirred for 12 h under these conditions. Yields given are isolated yields. [a]The reaction was carried out on a 1.5 mmol scale.

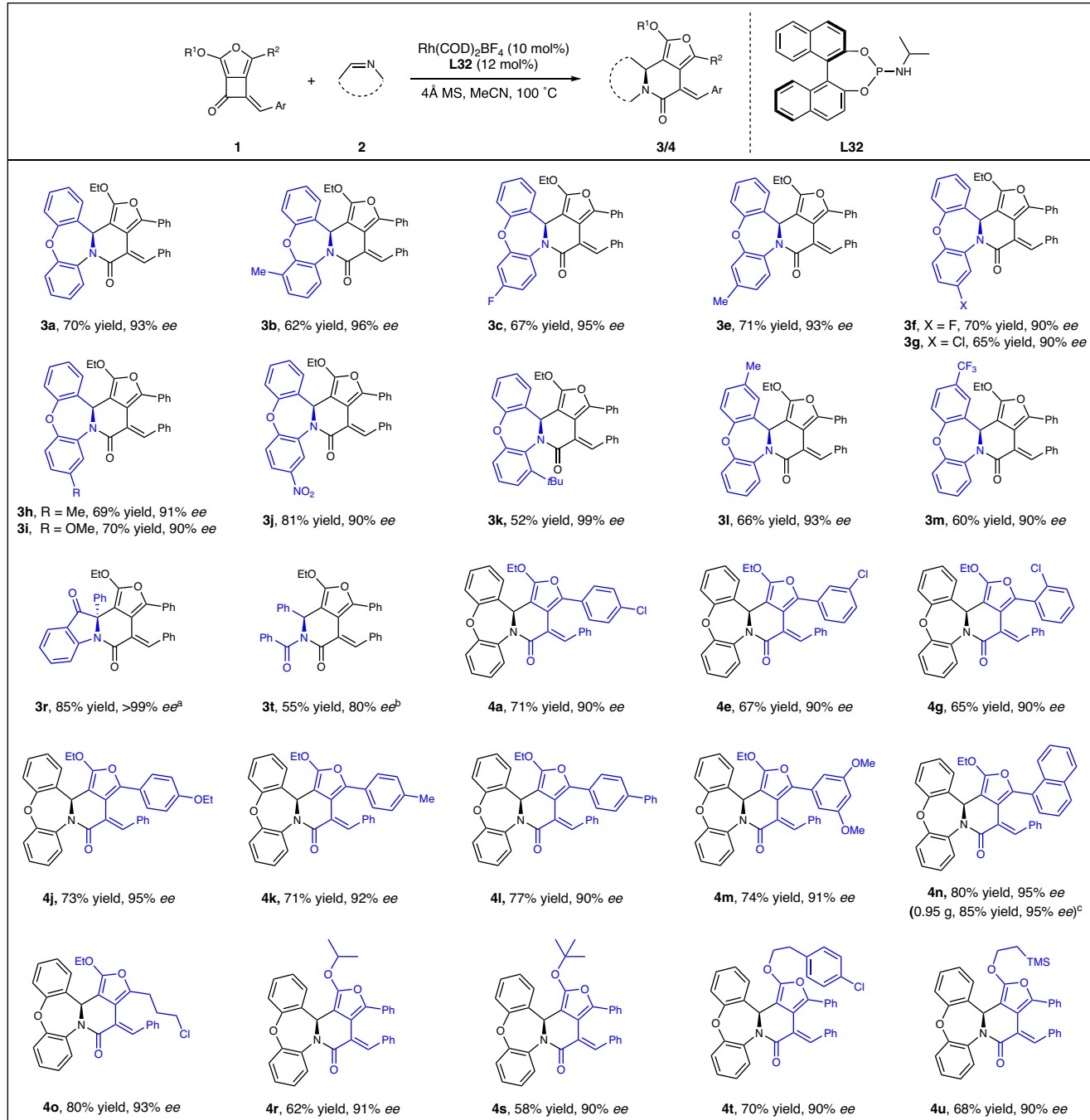

**Fig. 4 | Substrate scope for the enantioselective [4 + 2]-cycloaddition.** The reactions were performed on a 0.1 mmol scale: FCBs **1** (0.12 mmol, 1.2 equiv.), imines **2** (0.1 mmol, 1.0 equiv.), Rh(COD)$_2$BF$_4$ (4.1 mg, 10 mol%), 4 Å MS (50 mg), and chiral ligand **L32** (4.5 mg, 12 mol%) in MeCN (2.0 mL) under argon atmosphere at 100 °C for 12 h. Yields given are isolated yields. The enantiomeric excess was determined by chiral HPLC analysis. [a]Ligand **L2** was used instead of **L32**. [b]Ligand **L41** was used instead of **L32**. [c]The reaction was carried out on a 2.0 mmol scale.

Furthermore, the asymmetric [4 + 2]- and [4 + 4]-cycloaddition reactions were investigated with a variety of chiral gold complexes. Unfortunately, only moderate stereoselectivity was observed after extensive survey of the chiral ligand and counterions, although high yields were obtained in some cases (see Supplementary Tables 2, 5, and 6 in the Supplementary Information for details), suggesting a significant background reaction. On the other hand, due to the linear coordination geometry of the gold complex, the reaction generally led to moderate stereoselectivity[73,74]. However, when we used chiral rhodium complexes, the [4 + 2]-cycloaddition was achieved with high enantioselective control under the optimized conditions (10 mol% of Rh(COD)$_2$BF$_4$, 12 mol% of **L32**, in MeCN at 100 °C) (see Supplementary Tables 3 and 4 in SI for details), highlighting the ligand's boosting effect on the reactivity against the background reaction in the Rh-catalysis system. However, these conditions were not effective for the [4 + 4]-cycloaddition, which might be due to the inhibition effect of anthranil to the rhodium complexes. Under the optimal conditions, all the tested FCBs **1** and imines **2** underwent the [4 + 2]-cycloaddition reaction very well, yielding the chiral furan-fused six-membered lactams products **3a**–**4u** in good yields and excellent enantioselectivity (Fig. 4, up to 81%, 90–99% *ee*). Regardless of the electron-donating or electron-withdrawing nature of the substituents on different positions,

**Fig. 5 | Synthetic applications. a** Diels-Alder reaction of **4n. b** Hydrolysis reaction of **4n. c** Reduction reaction of **4n. d** Oxidation reaction of **4n. e** Diels-Alder reaction of **6a. f** Hydrolysis reaction of **6a. g** Reduction reaction of **6a. h** Oxidation reaction of **6a.**

including halide, trifluoromethyl, nitro, methoxy, methyl, and *tert*-butyl groups, these substituted imines all resulted in excellent outcomes in this reaction (**3a–3m**). Although further condition optimization is needed for other types of imines, [4 + 2]-cycloaddition product **3r** could be obtained in 85% yield as a single enantiomer when **L2** was used as the chiral ligand instead of **L32**; while, linear imine also works well, leading to analogous product **3t** in 55% yield with 80% ee when **L41** was used as the chiral ligand. Beyond the aryl substituted FCBs **1**, which provided the lactams **4a–4m** in high yields with 90–99% ee. These with alkyl chloride imbedded and a variety of alcohols derived ones were also well tolerated under the current conditions, leading to the chiral products **4o–4u** with comparable high reactivity and above 90% ee. In addition, the reaction could be performed on a 2.0 mmol scale with negligible effect on the yield and enantioselectivity (**4n**, 0.95 g, 85% yield, and 95% ee. Fig. 4, note d).

The significant advantage of this protocol is that the resultant poly-functionalized fused molecules can be readily transformed into various heterocycles with structural diversity by routine manipulations (Fig. 5). Treatment of **4n** with aryne precursor, 2-(trimethylsilyl)phenyl triflate, in the presence of CsF in MeCN led to the Diels−Alder reaction product **8** as a single diastereomer in 82% yield and 95% *ee*. The absolute configuration of **8** was confirmed by single-crystal X-ray diffraction analysis. Hydrolysis of **4n** occurred smoothly to produce the chiral butenolide product **9** in 91% yield with 92% *ee* and >20:1 *dr*. The absolute stereochemistry of **9** was determined as (*S*) using *X*-ray crystallographic analysis, and the stereochemistry of the chiral lactams **3** and **4** were assigned analogously. Reduction of the styryl motif with LiAlH₄ produced the hydroxy product **10** as a single diastereomer in 80% yield with 92% *ee*. The oxidative ring-opening reaction of **4n** occurred smoothly in the presence of *meta*-chloroperoxybenzoic acid (*m*-CPBA), giving **11** in 86% yield with 95% *ee* (see Supplementary Fig. 1 in SI for the determination of the relative configuration via 1d-noe

analysis). In addition, the furan-fused eight-membered lactam **6a** can also be converted into diverse *O*-bridged eight-membered lactams (**12–15**) in good yields under similar conditions. The relative configuration of **13** was confirmed by 1d-noe analysis (see Supplementary Fig. 2 in SI for detail); while the structure of **14** was established by X-ray crystallographic analysis.

Moreover, a few of synthesized compounds have been selected (**6b–6o, 6s, 7b–7c, 7e, 7g–7i, and 7l–7m**) for the anticancer activity evaluation on cell viability in comparison to the Paclitaxel via the CCK8 assay for MCF-7 (breast cancer), HCT-116 (colon cancer), A549 (lung adenocarcinoma), and KYSE-520 (esophageal squamous cell carcinoma) human cancer cell lines (see Supplementary Tables 12 and 13 and Supplementary Figs. 10–14 in the Supplementary Information for details). The results show that compounds **6e** and **6f** exhibited significant anticancer potency on HCT-116 (IC₅₀ = 0.50 ± 0.05 μM) and KYSE-520 (IC₅₀ = 0.89 ± 0.13 μM) human cancer cell lines, respectively (Fig. 6). Notably, the IC₅₀ of these two compounds against human normal colon mucosal epithelial cell lines (NCM 460 cells) is above 23 μM, which is more than tenfold value compared to the tested results with human cancer cell lines (see Supplementary Fig. 15 in SI for detail). Further structure-activity relationship study is ongoing in the laboratory.

To shed light on the mechanism of the developed cycloadditions using FCBs as the C4 synthon, control experiments of metal complexes with FCB **1b** were conducted in deuterated solvent to investigate C−C bond activation process (Fig. 7). The reaction of **1b** in the presence of Rh(COD)₂BF₄ (1.0 equiv.) was carried out in CD₃CN at 100 °C, which was intended to reveal the C−C bond insertion intermediate. Complete conversion of **1b** to the metallacycle species **A** takes place in 1.0 h, as is apparent from its ¹³C NMR spectrum (Fig. 7a)[75]. Moreover, an analogous phenomenon was observed when **1b** was treated with gold complex in CDCl₃, leading to the ¹³C NMR signal of the carbonyl group shift from 174.7 ppm to 194.3 ppm (Fig. 7b). These results clearly

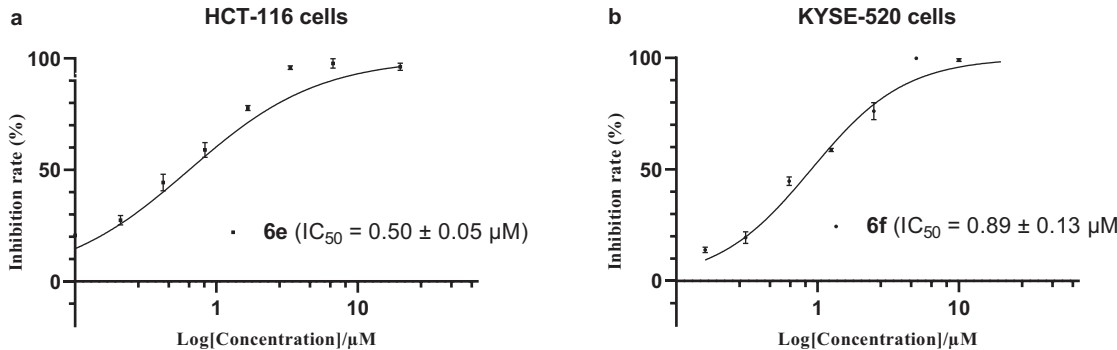

**Fig. 6 | The cytotoxicity of the synthetic compounds on HCT-116 and KYSE-520 human cell lines. a** Compound **6e** on the inhibition of HCT-116 cells. **b** Compound **6f** on the inhibition of KYSE-520 cells. All data were presented as mean values ± SD, $n$ = 3.

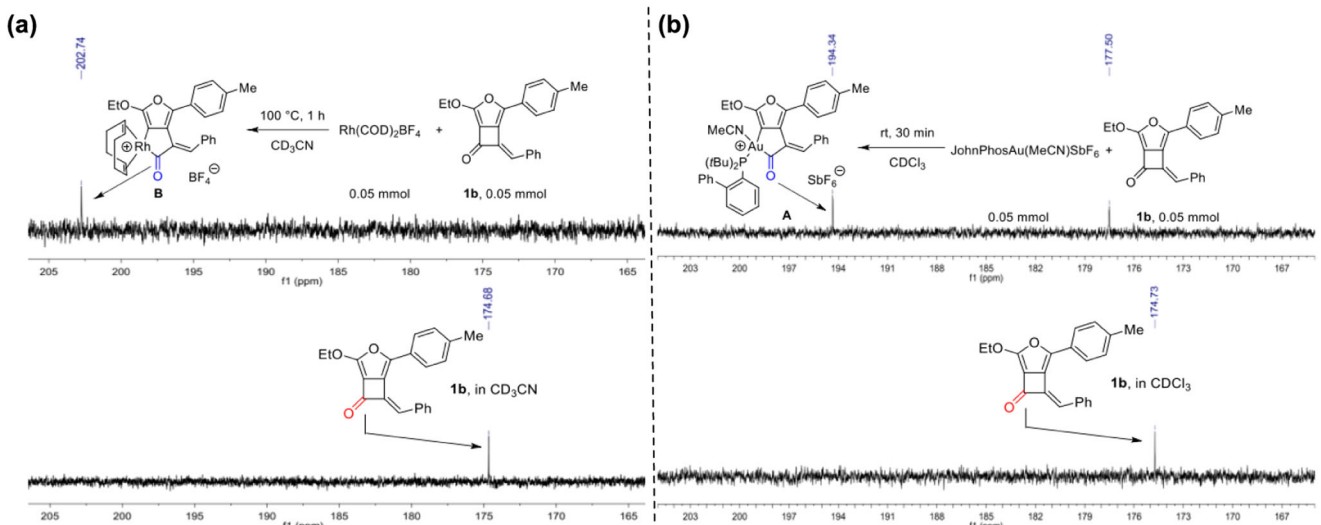

**Fig. 7 | Control experiments. a** ¹³C NMR spectra of **1b** and **1b** with Rh-complex in CD₃CN. **b** ¹³C NMR spectra of **1b** and **1b** with Au-complex in CDCl₃.

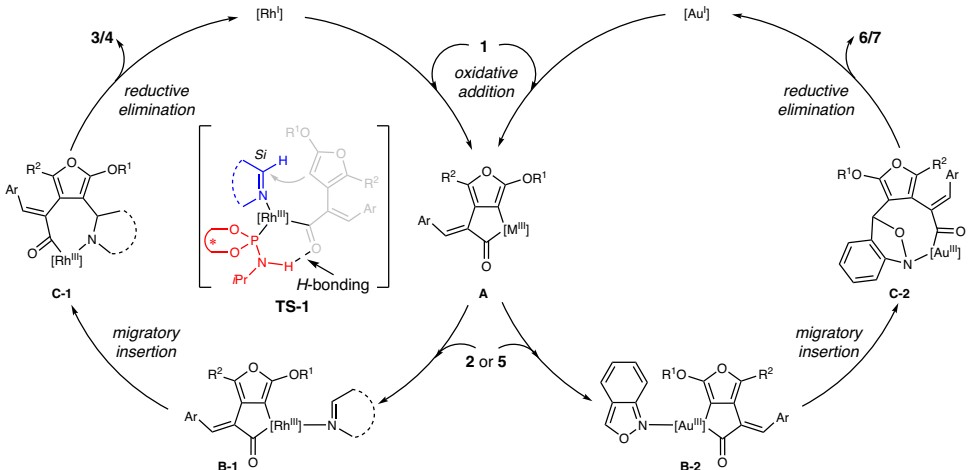

**Fig. 8 | Proposed reaction mechanism.**

suggested the initial C−C bond activation process in this cycloaddition reaction.

On the basis of above results and previous studies[43–59], a plausible reaction pathway has been proposed in Fig. 8. Initially, metallacycle **A** is generated from furan-fused cyclobutanone **1** through oxidative addition[75], while analogous C1−C7 bond cleavage could not occur in

this case[55], since the C1−C7 bond is imbedded in a α,β-unsaturated carbonyl motif, which requires much high bond cleavage energy then the formation of **A** due to the conjugation effect. Then, imine **2** or anthranil **5**[67] can coordinate with metal to form complexes **B-1** or **B-2**, followed by migratory insertion that leads to the corresponding seven-membered or nine-membered metallacycle species **C-1** or **C-2**. Finally,

reductive elimination gives the desired lactams products **3/4** or **6/7**, respectively. The exceptionally high enantioselectivity achieved by chiral ligand **L32** might be due to the potential *H*-bonding between the free N-H group and the carbonyl group of the substrate[76,77], since the *ee* decreased dramatically when analogous chiral ligand without a free N-H species was used (see Supplementary Table 3 in SI for detail: **L13**, 68% yield, 53% *ee*). A possible *Si*-face addition transient state **TS-1** has been proposed according to the obtained stereochemistry in the products.

In summary, we have developed a modular and atom-economic carboacylation reaction using furan-fused cyclobutanones as a versatile C4 building block via C–C bond activation. Using this method, the intermolecular highly enantioselective Rh(I)-catalyzed [4 + 2]-cycloaddition and Au(I)-catalyzed diastereoselective [4 + 4]-cycloaddition of the in situ formed metallacycle species has been established with imines and anthranils as corresponding dienophiles. This process produces unique poly-substituted furan-fused lactams in good yields and excellent stereoselectivity under mild conditions. Further synthetic derivatization of these structures led to densely functionalized six/eight-membered lactams with structural diversity in high yields. Meanwhile, a preliminary antitumor activity study of these generated products indicates that the eight-membered lactams **6** have high anticancer potency against human colon cancer cells (HCT-116) and esophageal squamous cell carcinoma cells (KYSE-520). Overcoming the catalytic protocol for the selective C–C bond activation of fused cyclobutanones should unlock the potential of these unique FCB scaffolds in heterocycle synthesis and other applications by enabling the design of analogous [4+n] annulation products. Further studies of this kind by expansion of the reaction scope to other dienophiles are ongoing in our laboratories, especially the asymmetric catalytic methods, and will be reported in due course.

## Methods

### Representative procedure for the [4 + 2] cycloaddition reaction
**Condition A**. To a 10-mL oven-dried vial containing a magnetic stirring bar, cyclobutanone **1** (0.12 mmol, 1.2 equiv.), imine **2** (0.1 mmol, 1.0 equiv.), IPrAuNTf$_2$ (4.3 mg, 5.0 mol%), and MeCN (2.0 mL) were added sequentially under argon atmosphere. After addition, the reaction mixture was stirred overnight at 40 °C until consumption of the material (monitored by TLC). The solvent was evaporated in vacuo. Then, the residues were purified by column chromatography on silica gel without any additional treatment (Hexanes: EtOAc = 50:1–10:1) to give the pure products **3** or **4** in good to high yields.

**Condition B**. To a 10-mL oven-dried vial containing a magnetic stirring bar, cyclobutanone **1** (0.12 mmol, 1.2 equiv.), imine **2** (0.1 mmol, 1.0 equiv.), Rh(COD)$_2$BF$_4$ (4.0 mg, 10 mol%), 4 Å MS (50 mg), chiral ligand **L32** (4.6 mg, 12 mol%), and MeCN (2.0 mL) were added sequentially under argon atmosphere at 100 °C. After addition, the reaction mixture was stirred overnight under these conditions until consumption of the material (monitored by TLC). The solvent was evaporated in vacuo. Then the residues were purified by column chromatography on silica gel without any additional treatment (Hexanes: EtOAc = 50:1–10:1) to give the pure products **3** or **4** in good to high yields with generally excellent enantioselectivity.

### Representative procedure for the [4 + 4] cycloaddition reaction
To a 10-mL oven-dried vial containing a magnetic stirring bar, cyclobutanone **1** (0.1 mmol, 1.0 equiv.), and anthranil **5** (0.11 mmol, 1.1 equiv.) in DCE (1.0 mL), was added a solution of JohnPhosAu(MeCN)SbF$_6$ (3.9 mg, 5.0 mol%) in DCE (1.0 mL) via syringe under argon atmosphere at 40 °C. After addition, the reaction mixture was stirred 12 h under these conditions until consumption of the material (monitored by TLC). The solvent was evaporated in vacuo. Then, the residues were purified by column chromatography on silica gel without

any additional treatment (Hexanes: EtOAc = 30:1–10:1) to give the pure products **6** or **7** in good to high yields.

### Reporting summary
Further information on research design is available in the Nature Portfolio Reporting Summary linked to this article.

## Data availability
All data are available from the corresponding author upon request. The data supporting the findings of this study are available within the paper and its Supplementary Information. Crystallographic data for the structures reported in this article have been deposited at the Cambridge Crystallographic Data Centre, under deposition numbers CCDC 2225032 (**4k**), 2218249 (**6f**), 2297136 (**8**), 2309002 (**9**), and 2348279 (**14**). Copies of the data can be obtained free of charge via https://www.ccdc.cam.ac.uk/structures/.

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

## Acknowledgements

Support for this research from the National Natural Science Foundation of China (22371309) and the Guangdong Provincial Key R&D Programme (21202107201900002) is greatly acknowledged. We also thank Prof. Michael P. Doyle from UTSA for the discussion and suggestions during the preparation of the manuscript.

## Author contributions

X.X. supervised the project and wrote the manuscript. K.H. and M.L. designed the experiments. K.H., M.L., L.Q., G.C., and M.B. performed the experiments and analyzed the data. X.J. and J.H. carried out the antic-ancer activity evaluation. All authors discussed the results and com-mented on the article.

## Competing interests

The authors declare no competing interests.
