## [Peer Review File · Nature Communications]

Catalytic [4+2]- and [4+4]-cycloaddition using furan-fused cyclobutanone as a privileged C4 synthonREVIEWER COMMENTS

Reviewer #1 (Remarks to the Author):

The development of novel synthons which could participate in cycloadditions, especially the high-order cycloadditions, enables rapid construction of skeletally different cyclic frameworks efficiently. In this manuscript, Xu and co-workers reported the utilization of furan-fused cyclobutanones (FCBs), originally developed by the same group, as novel C4 components in [4+2] and [4+4] cycloadditions with imines and anthranils under the catalysis of rhodium(I) and gold(I), respectively. Moreover, the asymmetric [4+2] cycloadditions were achieved by employing a chiral rhodium complex. In addition, preliminary antitumor activity studies of some products further highlighted the potential usefulness of the current method.

As a result, the authors developed a new type of synthons that are capable of engaging in diverse [4+n] cycloadditions to access intriguing heterocycles. However, the current protocol suffered from relatively narrow substrate scope and the mechanism was not well elucidated.

1) The structure of the furan-fused cyclobutanones is rather specific. Is it possible to test the relevant ones with more types of substitution patterns?

2) Besides the cyclic imines **2**, could the simple non-cyclic aldimines and ketimines be utilized in the [4+2] cycloadditions with FCBs?

3) In the asymmetric [4+2] cycloaddition reaction, an unusual phosphoramidite ligand L32 with a free N-H outperformed others in terms of enantiocontrol. More control experiments would be conducted to rationalize the key factors for stereo-induction. In addition, corresponding explanations and stereo-control mode would be provided.

4) As demonstrated in Table 4, good enantioselectivity was uniformly obtained for various dibenzazepine-based imines **2** in the [4+2] cycloadditions. But the results for products derived from imines with different backbones, including products **3p–3s**, were not provided.

5) The relative configurations of products **10**, **13** and **14** should be assigned by X-ray crystallographic analysis or other means. Moreover, the N-atom of product **6/7** probably be chiral because of its rigid conformation and previous studies have confirmed that (Angew. Chem. Int. Ed. 2020, 59, 10396–1040). Thus, the relative configuration of **6/7** should be provided.

6) Some other minor errors: typo in Figure 1B “Asymmetric catalysis”; please indicate the exact configuration of L33 in Table S3.

7) Both products **6** and **7** have a Michael acceptor motif. How about their activity against normal cell lines other than cancer ones?

This reviewer would like to support the publication on Nat. Commun. after finely addressing the issues raised above.

Reviewer #2 (Remarks to the Author):

In this manuscript, Xu and coworkers developed two types of cycloaddition reactions by using furan-fused cyclobutenones (FCBs) as C4 synthon via C-C bond activation. One was intermolecular [4+2]-cycloaddition with imines (with good to excellent enantioselectivity) and the other was intermolecular [4+4]-cycloaddition with anthranils. A series of intriguing furan-based heterocycles were synthesized using this method, from which compound **6e** and **6f** was found to have antitumor activity in preliminary study. Mechanistic studies suggested the initial C-C bond activation process in this cycloaddition.

I have to say the authors present us with solid work in this manuscript. Both of the two parts

provided plenty of examples in moderate to good yield for support. And what is even more interesting is that the enantioselectivity of the [4+2]-cycloaddition can be well-controlled. Therefore, I believe this would be acceptable for publication in Nature Communication if the authors could address the problems below.

(1) The Rh with chiral ligand could be used for highly enantioselective [4+2] cycloaddition, however, Au with some chiral ligands only gave moderate results. May the authors provide some explanations about this question in the manuscript? Maybe DFT calculation could help.

(2) The products shown in the manuscript are all formed based on the mechanism of C1-C2 cleavage. As far as I know, C1-C8 cleavage (of benzocyclobutenones) could also be achieved in some conditions (for example, *Angew. Chem. Int. Ed.* 2021, 60, 19079-19084.). I just wonder how C1-C2 regioselectivity could be controlled in this work, rather than C1-C7 cleavage.

(3) The title involving [4+n] should be revised, actually, there were [4+2] and [4+4] in current work. I would suggest the title could be "Catalytic [4+2] and [4+4]-cycloaddition using furan-fused cyclobutanone as a privileged C4 synthon"

(4) Minor errors. The color should be consistent with substrates in the Table 2. For example, the substrates 2 were blue, then they were changed into black in the scope of FCBs.

For Reviewer 1: The development of novel synthons which could participate in cycloadditions, especially the high-order cycloadditions, enables rapid construction of skeletally different cyclic frameworks efficiently. In this manuscripts, Xu and co-workers reported the utilization of furan-fused cyclobutanones (FCBs), originally developed by the same group, as novel C₄ components in [4+2] and [4+4] cycloadditions with imines and anthranils under the catalysis of rhodium(I) and gold(I), respectively. Moreover, the asymmetric [4+2] cycloadditions were achieved by employing a chiral rhodium complex. In addition, preliminary antitumor activity studies of some products further highlighted the potential usefulness of the current method.

As a result, the authors developed a new type of synthons that are capable of engaging in diverse [4+n] cycloadditions to access intriguing heterocycles. However, the current protocol suffered from relatively narrow substrate scope and the mechanism was not well elucidated.

We appreciate the reviewer for the comments. For the substrate scope, as mentioned by this reviewer, FCBs is a novel synthon which has not been used in cycloaddition reactions and other C-C bond activation transformations. Thus, novel synthetic methodologies could be envisioned using FCBs as practical and robust synthons, and our group are continuing work in this direction, including [4+2]-cycloaddition with alkynes and alkenes, [4+3]-cycloaddition with 1,3-dipoles, which will be submitted for publication in due course sequentially.

For the reaction mechanism part, we have conducted a control experiment to monitoring the key C-C bond activation step by ¹³C NMR spectra analysis of the crude reaction mixture (Fig. 4 in the manuscript), which indicated an obvious carbonyl signal shift that is consisting with precious study for the analogous C-C activation process (ref. 73 Angew. Chem. Int. Ed. 54, 5236–5240 (2015)). Once this reactive intermediate was formed, a variety of cycloaddition reactions could be envisioned with this metallacycle species as novel C₄ synthon based on the analogous chemistry of benzocyclobutenones (BCBs) (see ref 43-59 in the manuscript). So far, only two examples have been reported for the catalytic transformations of furan-fused cyclobutanones (FCBs), and more detailed mechanistic insights will be disclosed in due course with the development of this chemistry.

1) The structure of the furan-fused cyclobutanones is rather specific. Is it possible to test the relevant ones with more types of substitution patterns?

We appreciate the reviewer for the suggestions. The substitution pattern of furan-fused cyclobutanones is quite broad, a variety of substitutions, aryl or alkyl groups, could be incorporated on the different positions of FCB scaffolds (see our previous work for the synthesis of FCBs for detail: ref 66 Nat. Commun. 14, 6378 (2023). Moreover, for the ether part, different alcohols derived diazo compounds could smoothly converted to the FCBs. Especially, the homopropargyl alcohol derived one could be converted to the fused tricyclic system under rhodium (I) catalysis (see below). We are still working on this project, and will report this result soon.

2) Besides the cyclic imines **2**, could the simple non-cyclic aldimines and ketimines be utilized in the [4+2] cycloadditions with FCBs?

*We appreciate the reviewer for the suggestions. We are still working on this direction. For the reaction with ketimine **2r**, desired product **3r** was obtained in 85% yield with >99% ee under the currently optimized conditions. These results have been added to the revised manuscript with discussion, and corresponding data have been added to the SI.*

Table S1 | Condition optimization for the Rh-catalyzed enantioselective [4+2]-cycloaddition with imine **2r^{a,b,c}**

^aReaction conditions: **1a** (34.8 mg, 0.11 mmol, 1.1 equiv.), **2r** (20.8 mg, 0.1 mmol), Rh(COD)₂BF₄ (2.0 mg, 5.0 mol%), chiral ligand (6.0 mol%) in MeCN (2.0 mL) under argon atmosphere at 100 °C for 12 h. ^bDetermined by ¹H NMR analysis of the crude reaction mixture based on internal standard (1,3,5-trimethoxybenzene). ^cDetermined by chiral HPLC analysis with a Chiralpak IC column. NR = no reaction.

*For the [4+2]-cycloaddition with non-cyclic aldimine **2t**, the desired product **3t** was obtained in 55% yield with 80% ee under the currently optimized conditions. These results have been added to the revised manuscript with discussion, and corresponding data have been added to the SI.*

Table S2 | Condition optimization for the Rh-catalyzed enantioselective [4+2]-cycloaddition with non-cyclic aldimine **2t^a**

Entry	Ligand (6.0 mol%)	Solvent	T (°C)	Yield (%) ^b	ee (%) ^c
1	L2	MeCN	100	60	45
2	L3	MeCN	100	68	66
3	L5	MeCN	100	62	10
4	L6	MeCN	100	57	65
5	L32	MeCN	100	58	15
6	L41	MeCN	100	55	80
7	L41	DCE	100	57	50
8	L41	PhMe	100	61	4
9	L41	MeCN	60	NR ^d	-
10	L41	MeCN	80	<5 ^d	-
11	L44	MeCN	100	43	56
12	L45	MeCN	100	NR ^d	-

^aReaction conditions: **1a** (34.8 mg, 0.11 mmol, 1.1 equiv.), **2t** (20.8 mg, 0.1 mmol), $\text{Rh}(\text{COD})_2\text{BF}_4$ (2.0 mg, 5.0 mol%), **L** (6.0 mol%) in 2.0 mL solvent at 100 °C under argon atmosphere for 12 h.

^bDetermined by ¹H NMR analysis of the crude reaction mixture based on internal standard (1,3,5-trimethoxybenzene). ^cDetermined by chiral HPLC analysis with a Chiralpak IC column. ^dmost of starting material **1a** was recovered.

3) In the asymmetric [4+2] cycloaddition reaction, an unusual phosphoramidite ligand L32 with a free N-H outperformed others in terms of enantiocontrol. More control experiments would be conducted to rationalize the key factors for stereo-induction. In addition, corresponding explanations and stereo-control mode would be provided.

We appreciate the reviewer for the suggestions. We have done the comparison experiment with non-free N-H ligand L13, which gave the product in comparable high yield but in lower 53% ee. These results implied a potential H-bonding between the ligand and substrate. Corresponding discussion have been added to the revised manuscript with new references (ref 76 and 77: Angew. Chem. Int. Ed. 48, 2162–2165 (2009); Chem. Sci. 9, 1919–1924 (2018)). Moreover, stereo-control mode via a Si-face addition transient state has been proposed in the revised manuscript with discussion according to the obtained stereochemistry in the products.

4) As demonstrated in Table 4, good enantioselectivity was uniformly obtained for various dibenzazepine-based imines **2** in the [4+2] cycloadditions. But the results for products derived from imines with different backbones, including products **3p–3s**, were not provided.

*We appreciate the reviewer for the suggestions. Further optimization has been conducted for these imines. For ketimine **2r**, desired product **3r** was obtained in 85% yield with >99% ee under the currently optimized conditions using L2 as chiral ligand; and for non-cyclic aldimine **2t**, the desired product **3t** was obtained in 55% yield with 80% ee under the currently optimized conditions using L41 as chiral ligand (see the response to the second question of this reviewer). These results have been added to the revised manuscript with discussion, and corresponding data have been added to the SI.*

For the other types of imines, the reaction conditions are still under optimization, and hope could be submitted for publication in due course after further optimization (see the listed results below for details).

Table S3 | Condition optimization for the Au-catalyzed enantioselective [4+2]-cycloaddition with **2p^a**

Entry	[M] (x mol%)	$AgNTf_2$ (y mol%)	Yield (%) ^b	ee (%) ^c
1	L2 [AuCl] ₂ (5.0)	10	41	40
2	L3 [AuCl] ₂ (5.0)	10	36	13
3	L5 [AuCl] ₂ (5.0)	10	57	20
4	L6 [AuCl] ₂ (5.0)	10	42	16
5	L8 AuCl (5.0)	5.0	<5 ^d	-
6	L35 [AuCl] ₂ (5.0)	10	35	32
7	L36 [AuCl] ₂ (5.0)	10	59	11
8	L46 [AuCl] ₂ (5.0)	10	47	53

^aReaction conditions: **1a** (34.8 mg, 0.11 mmol, 1.1 equiv.), **2p** (17.9 mg, 0.1 mmol), chiral gold catalyst (5.0 mol%), $AgNTf_2$ (10 or 5.0 mol%) in MeCN (2.0 mL) under argon atmosphere at 40 °C for 12 h. ^bDetermined by ¹H NMR analysis of the crude reaction mixture using mesitylene as an internal standard. ^cDetermined by chiral HPLC analysis with Chiralpak IC column. ^dMost of the starting materials **1a** and **2p** were recovered.

Table S4 | Condition optimization for the catalytic [4+2]-cycloaddition with 2q^a

Entry	[M] (x mol%)	AgNTf ₂ (y mol%)	Yield (%) ^b	ee (%) ^c
1	Rh(COD) ₂ BF ₄ (5.0)	-	ND ^d	-
2	Cu(MeCN) ₄ PF ₆ (5.0)	-	23	-
3	[Ir(COD)Cl] ₂ (5.0)	-	12	-
4	AgOTf (5.0)	-	42	-
5	Zn(OTf) ₂ (10)	-	61	-
6	IPrAuNTf ₂ (5.0)	-	70	-
7	L2 [AuCl] ₂ (5.0)	10	39	<5
8	L3 [AuCl] ₂ (5.0)	10	42	<5
9	L35 [AuCl] ₂ (5.0)	10	12	<5
10	L5 [AuCl] ₂ (5.0)	10	37	<5
11	L6 [AuCl] ₂ (5.0)	10	31	6
12	L8 AuCl (5.0)	5.0	16	5
13	L36 [AuCl] ₂ (5.0)	10	43	<5
14	L46 [AuCl] ₂ (5.0)	10	27	<5

^aReaction conditions: **1a** (34.8 mg, 0.11 mmol, 1.1 equiv.), **2q** (14.6 mg, 0.1 mmol), catalyst (x mol%) in MeCN (2.0 mL) under argon atmosphere at 40 °C for 12 h. ^bDetermined by ¹H NMR analysis of the crude reaction mixture using mesitylene as an internal standard. ^cDetermined by chiral HPLC analysis with Chiralpak IC column. ^dThis reaction was carried out in 100 °C. ND = no detected.

Table S5 | Condition optimization for the catalytic [4+2]-cycloaddition with 2s^a
Entry	[M] (5.0 mol%)	Solvent	T (°C)	Yield (%) ^b	ee (%) ^c
1	Rh(COD) ₂ BF ₄	MeCN	100	ND ^d	-
2	IPrAuNTf ₂	MeCN	40	40	-
3	Cu(MeCN) ₄ PF ₆	MeCN	40	21	-
4	Cu(MeCN) ₄ PF ₆	DCE	40	30	-
5	Cu(MeCN) ₄ PF ₆ / L47 (6.0)	DCE	40	20	<5
6	Cu(MeCN) ₄ PF ₆ / L48 (6.0)	DCE	40	NR	-
7	Cu(MeCN) ₄ PF ₆ / L49 (6.0)	DCE	40	18	<5

^aReaction conditions: **1a** (34.8 mg, 0.11 mmol, 1.1 equiv.), **2s** (19.5 mg, 0.1 mmol), catalyst (5.0 mol%) in 2.0 mL solvent under argon atmosphere at 40 °C for 12 h. ^bDetermined by ¹H NMR analysis of the crude reaction mixture using mesitylene as an internal standard. ^cDetermined by chiral HPLC analysis with Chiralpak IC column. ^dMost of starting material **2s** was recovered. ND = no detected. NR = no reaction.

5) The relative configurations of products 10, 13 and 14 should be assigned by X-ray crystallographic analysis or other means. Moreover, the N-atom of product 6/7 probably be chiral because of its rigid conformation and previous studies have confirmed that (Angew. Chem. Int. Ed. 2020, 59, 10396–1040). Thus, the relative configuration of 6/7 should be provided.

We appreciate the reviewer for the suggestions. The relative configurations of products 10 and 13 were confirmed by 1d-noe NMR analysis. And the relative configurations of product 14 was confirmed by X-ray crystallographic analysis. Corresponding revision has been made the manuscript and these data have been

added to the SI. For products 6/7, only racemic version has been reported so far, thus, no change has been made, and corresponding reference has been cited the manuscript as ref 67: Angew. Chem. Int. Ed. 59, 10396–10400 (2020).

6) Some other minor errors: typo in Figure 1B “Asmmetric catalysis”; please indicate the exact configuration of L33 in Table S3.

Thank you for pointing this out. The mistake has been corrected. The ligand L33 was prepared with racemic α -phenylethylamine, thus, no change has been made.

7) Both products 6 and 7 have a Michael acceptor motif. How about their activity again normal cell lines other than cancer ones? This reviewer would like to support the publication on Nat. Commun. after finely addressing the issues raised above.

We appreciate the reviewer for the suggestions. We have tested the products 6e and 6f on human normal colon mucosal epithelial cell lines (NCM 460 cells), which shows more than tenfold IC_{50} value in compassion to the results on HCT-116 ($IC_{50} = 0.50 \pm 0.05 \mu M$) and KYSE-520 ($IC_{50} = 0.89 \pm 0.13 \mu M$) human cancer cell lines. These results have been discussed in the revised manuscript, and corresponding data have been added to the SI.

For Reviewer 2: In this manuscript, Xu and coworkers developed two types of cycloaddition reactions by using furan-fused cyclobutenones (FCBs) as C4 synthon via C-C bond activation. One was intermolecular [4+2]-cycloaddition with imines (with good to excellent enantioselectivity) and the other was intermolecular [4+4]-

cycloaddition with anthranils. A series of intriguing furan-based heterocycles were synthesized using this method, from which compound 6e and 6f was found to have antitumor activity in preliminary study. Mechanistic studies suggested the initial C-C bond activation process in this cycloaddition.

I have to say the authors present us with solid work in this manuscript. Both of the two parts provided plenty of examples in moderate to good yield for support. And what is even more interesting is that the enantioselectivity of the [4+2]-cycloaddition can be well-controlled. Therefore, I believe this would be acceptable for publication in Nature Communication if the authors could address the problems below.

We appreciate the reviewer for the comments.

(1) The Rh with chiral ligand could be used for highly enantioselective [4+2] cycloaddition, however, Au with some chiral ligands only gave moderate results. May the authors provide some explanations about this question in the manuscript? Maybe DFT calculation could help.

We appreciate the reviewer for the suggestions. This may due to the two major reasons, the significant background reaction with non-chiral gold catalyst, and the linear coordination geometry of the gold complex, which generally led to moderate stereoselectivity. Corresponding comment has been added to the revised manuscript with references 74 and 75:

74. Zi, W. & Toste, F. D. Recent advances in enantioselective gold catalysis. *Chem. Soc. Rev.* **45**, 4567–4589 (2016)

75. Zuccarello, G., Mayans, J. G., Escofet, I., Scharnagel, D., Kirillova, M. S., Pérez-Jimeno, A. H., Calleja, P., Boothe, J. R. & Echavarre, A. M. Enantioselective folding of enynes by gold(I) catalysts with a remote C_2 -chiral element. *J. Am. Chem. Soc.* **141**, 11858-11863 (2019)

(2) The products shown in the manuscript are all formed based on the mechanism of C1-C2 cleavage. As far as I know, C1-C8 cleavage (of benzocyclobutenones) could also be achieved in some conditions (for example, *Angew. Chem. Int. Ed.* 2021, 60, 19079-19084.). I just wonder how C1-C2 regioselectivity could be controlled in this work, rather than C1-C7 cleavage.

*We appreciate the reviewer for the comments. However, C1-C7 bond cleavage could not occur in our case under current conditions, since the C1-C7 bond is imbedded in a α,β -unsaturated carbonyl motif, which requires much high bond cleavage energy than the C1-C2 cleavage due to the conjugation effect. A brief comment has been added to the revised manuscript, and the reference has been cited as ref 55: Angew. Chem. Int. Ed. **60**, 19079-19084 (2021).*

(3) The title involving [4+n] should be revised, actually, there were [4+2] and [4+4] in current work. I would suggest the title could be “Catalytic [4+2] and [4+4]-cycloaddition using furan-fused cyclobutanone as a privileged C4 synthon”

We appreciate the reviewer for the suggestion, and corresponding revision has been made as requested.

(4) Minor errors. The color should be consistent with substrates in the Table 2. For example, the substrates 2 were blue, then they were changed into black in the scope of FCBs.

We appreciate the reviewer for the suggestion, and corresponding revision has been made as requested.

REVIEWERS' COMMENTS

Reviewer #1 (Remarks to the Author):

In the revised manuscript, the authors have well addressed the issues raised by reviewers. The substrate scope, especially for imine-based ones, has been expanded properly. The current work showcases the rational design of new synthons to construct potentially valuable products, even enantioselectively. As a result, this reviewer would like to support the publication as is stands.

Reviewer #2 (Remarks to the Author):

I have carefully checked the revised manuscript and the supporting informations. All the issues raised by the reviewers have been well addressed. It can be accepted as it is.